# Polyphenols and Small Phenolic Acids as Cellular Metabolic Regulators

**Mark Obrenovich** [1,2,3,4,5,6,*], **Yi Li** [7,*], **Moncef Tayahi** [3,8] **and V. Prakash Reddy** [9]

1    Department of Veteran's Affairs Medical Center, Research Service, Cleveland, OH 44106, USA
2    Department of Chemistry, Case Western Reserve University, Cleveland, OH 44106, USA
3    The Gilgamesh Foundation for Medical Science and Research, Cleveland, OH 44116, USA
4    Department of Medicinal and Biological Chemistry, College of Pharmacy and Pharmaceutical Sciences, University of Toledo, Toledo, OH 43614, USA
5    Departments of Chemistry, Cleveland State University, Cleveland, OH 44115, USA
6.   Biological and Environmental Sciences, Cleveland State University, Cleveland, OH 44115, USA
7    Department of Nutrition and Dietetics, Saint Louis University, Saint Louis, MO 63103, USA
8    Department Electrical and Computer Engineering, CEAS College of Engineering, University of Cincinnati, Cincinnati, OH 45221, USA
9    Department of Chemistry, Missouri University of Science and Technology, Rolla, MO 65409, USA
*    Correspondence: meo5@case.edu (M.O.); yi.li@health.slu.edu (Y.L.)

**Abstract:** Polyphenols and representative small phenolic acids and molecules derived from larger constituents are dietary antioxidants from fruits, vegetables and largely other plant-based sources that have ability to scavenge free radicals. What is often neglected in polyphenol metabolism is bioavailability and the role of the gut microbiota (GMB), which has an essential role in health and disease and participates in co-metabolism with the host. The composition of the gut microbiota is in constant flux and is modified by multiple intrinsic and extrinsic factors, including antibiotics. Dietary or other factors are key modulators of the host gut milieu. In this review, we explore the role of polyphenols and select phenolic compounds as metabolic or intrinsic biochemistry regulators and explore this relationship in the context of the microbiota–gut–target organ axis in health and disease.

**Keywords:** polyphenols; microbiota; small phenolic acids; mass spectrometry; green tea; resveratrol; catechins; eugenol; whole genome sequencing

## 1. Introduction

Polyphenolic antioxidants from dietary sources are frequently a topic of interest, largely due to widespread scientific agreement and increased findings that they may help lower the incidence of certain diseases, such as cancers, heart and cardiovascular disease, type 2 diabetes (T2D) and neurodegenerative diseases. While the mechanisms remain multifactorial, they include properties affecting nucleic acid, lipid and protein damage, such as non-enzymatic glycation or gloxidative stress, and even may have antiaging properties [1]. Polyphenols are effective for a number of diseases such as those stemming from inflammation, oxidative stress [2], pyretic activity [3], reproductive disorders, nervous system disorders, elevated blood glucose [4], microbial and viral infections, cholesterol irregularities, cellular proliferation and tumorigenesis, hypertension, pain management and digestive complications [1]. Neurodegenerative diseases, which exhibit distinct etiologies, are largely protracted and constitute constant deterioration in neuronal function, neuronal loss, brain atrophy and memory impairment. Oxidative stress and inflammation are common in Alzheimer's disease (AD), Parkinson's disease (PD), Huntington's disease and amyotrophic lateral sclerosis [5], and many other diseases [6].

Dietary polyphenolic antioxidants are frequently a topic of interest due to widespread scientific agreement they can help lower the incidence of certain cancers, cardiovascular and neurodegenerative diseases and DNA damage, and may have possible anti-aging properties. Many studies have indicated that polyphenols have protective effects against chronic diseases, including obesity, type 2 diabetes, cardiovascular disease, neurodegenerative diseases and some types of cancer [7]. Moreover, the increased risk for these age-related diseases, along with the decreased cellular functions or improper gain of function involving damage to cellular macromolecules, such as lipids, proteins and nucleic acid, are associated with the hallmarks of aging. These include epigenetic changes, genomic instability, altered intercellular communication, loss of proteostasis, mitochondria dysfunction, deregulated nutrient sensing, stem cell exhaustion, cellular senescence and apoptosis and more [8]. Polyphenols can attenuate these hallmarks of disease, in part, by preventing damage to by exogenous and endogenous cellular stressors including oxidative stress, inflammatory stress, endoplasmic reticulum stress and reduce innate ability of cellular recovery. The health benefits of polyphenol are related to bioavailability, which partly depends on industrial and domestic food processing and digestibility in the gastrointestinal tract. Moreover, cellular metabolism of the food components and the co-metabolism with the microbiota can improve the uptake of these components, but do not necessarily improve parent compound bioavailability. Efforts to modulate, as proof of principle, that oxidative stress and concomitant inflammation, demonstrate modulation of oxidative stress, especially by using dietary mitochondria augmenting antioxidants. This represents a promising approach to prevent or treat diseases, such as Parkinson's disease and Loue Gehrig's disease (amyotrophic lateral sclerosis) [5,9]. However, most mitochondria-targeted antioxidants with beneficial effects, in associated models, have often failed to demonstrate clinical therapeutic benefit. Several questions remain to be clarified, such as the role of the microbiota–gut–brain axis in these poor outcomes, and little has been done to consider the microbial species in these study subjects [1,5,10]. Any role played by oxidative stress in neurodegeneration, such as Parkinson's and Alzheimer's disease pathogenesis, emphasize mitochondria as generators of deleterious reactive oxygen species (ROS) and reactive nitrogen species (RNS), which are both targets for oxidative stress-related pathobiological mechanisms. Polyphenols and phenolic compounds, such as catechins, quercetin and curcuminoids, are actively researched modulators of PD and hold promise in treating or preventing several neurodegenerative diseases [9].

## 2. Bioavailability of Dietary Polyphenols

Polyphenols and phenolic acid food components must be bioavailable to exert any so-called biologic effects. That said, the medicinal potential of small phenolic acids and the precursor polyphenols, such as curcumin (*Curcuma longa*) and green tea (*Camellia sinensis)*, and its catechins can be severely affected by limited systemic and target tissue bioavailability and rate of metabolism. Many of these beneficial compounds only reach the systemic circulation though limited gut absorbance [1,9,11]. Nevertheless, the health benefits of polyphenols are related to bioavailability, which depends on the food processing as well as digestion in the gastric intestine and subsequent bacterial metabolism in the gastric intestine and cellular metabolism of these food components. Data regarding polyphenol absorption and tissue distribution are derived largely from animal studies [1,9,11].

Green tea is a particularly attractive candidate to study and trace bio-transformations by the gut microbiota [10,12]. In this regard, composition and metabolism have been well characterized by the four main tea catechins: (-)-epigallocatechin-3-*O*-gallate (EGCG), (-)-epicatechin-3-*O*-gallate (ECG), (-)-epigallocatechin (EGC), (-)-epicatechin (EC) (See Figure 1) account for 92% of all the flavonoids found in brewed green tea according to the Flavonoid Content of Selected Foods in the USDA Database [1]. Most of these catechins are also known as tannins, which are largely secondary plant metabolites that were originally discovered because of a strong interaction with collagen both in situ and in vivo. They are

used frequently in the so-called "tanning" process, which converts animal hides to leather. Tannins are defined as high-molecular-weight polyphenol polymers that can precipitate proteins from solution. Key to their importance is the bioavailability of lack thereof of these compounds in vivo, ultimately depends on their degree of polymerization. The difficulty in bioavailability with oligomeric forms of these compounds were found to be variant and the more bioavailable they are the higher the demonstrated high antioxidant activity when compared to their monomeric or polymeric forms. These were found to be, in the order of radical scavenging effectiveness, as follows: ECG > EGCG > EGC > EC > catechin (See Figure 1).

Similarly, proanthocyanidines undergo depolymerizations generally by thiolysis, which involves cleavage of interflavonoid linkages and then nucleophilic attack by thiol groups largely at the C4 position on the molecule [13]. Depolymerization also has been reported using other nucleophiles such as L-cysteine and phloroglucinol [1].

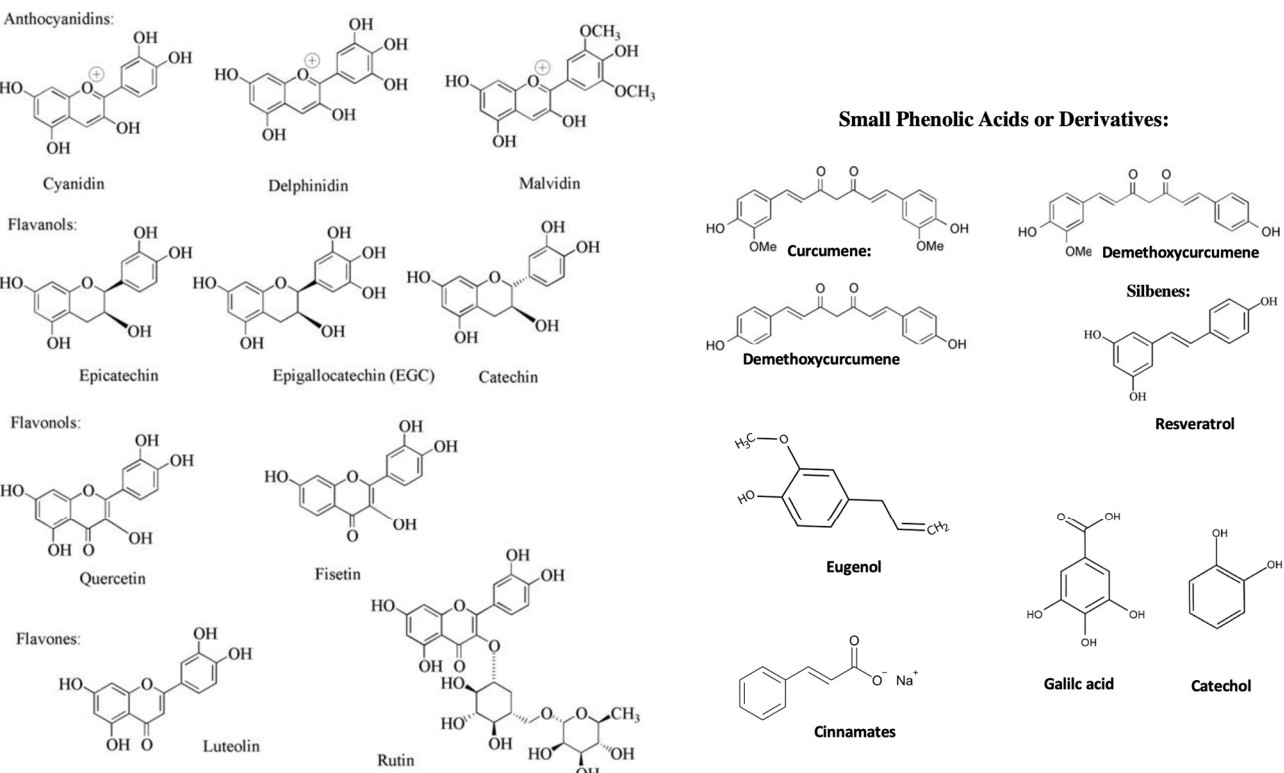

**Figure 1.** Structures of select polyphenols and phenolic acids discussed in this review.

## 3. Bacterial and Host Co-metabolism of Polyphenols and Phenolic Acid Metabolites

A vegan diet (VD) is known to demonstrate beneficial health effects; however, the role of the gut microbiota is unclear and largely underexplored. What we know is that flavonoids constitute the largest group of dietary polyphenols [14], which despite their heterogeneity, are metabolized through a limited series of common metabolic steps in part mediated by the MG [15,16]. However, in order for phenolic acids to have their affects they must be processed by the microbiota-gut, which we and other have established [17]. In that regard, a 4-week dietary study, vegan diet vs. meat-rich diet (MD) was conducted in a monocentric, randomized, controlled trial with a parallel group. Here, fecal samples from 53 healthy, omnivore, normal-weight participants (62% female, mean 31 years of age), were collected pre- and post-trial, and were analyzed using 16S rRNA gene amplicon sequencing [18]. These authors showed that alpha and beta diversity did not differ significantly between MD and VD subjects. Baseline and end samples emphasized a highly intra-individual microbial composition, which unlike animals housed together are much

more diverse but similar to human gut microbial content. In that regard, the overall gut microbiota phyla were not remarkably altered between VD and MD after the trial. These same authors noted *Coprococcus* was found to be increased in VD, whereas it was decreased in MD. *Faecalibacterium* and *Roseburia* were increased in MD whereas they were decreased in VD. In MD, the signatures of *Bacteroides*, *Faecalibacterium*, *Clostridium* sp. and *Roseburia* were enriched and in VD were depleted after the trial. Moreover, multiple amplicon sequence variants of genera *Bacteroides*, *Blautia*, *Dialister*, *Faecalibacterium*, and *Ruminococcus*, however, were depleted in MD but enriched in VD after the trial [14].

The previous work and unpublished observations in *Clostridioides difficile* patient populations led to the hypotheses around mechanisms of pathogenesis, which suggests that *Clostridioides difficile* and its course of infection may be affected by altering the diet, particularly by omitting meat, when actively affected. Perhaps this suggests that adding polyphenols to the diet could be tried to lessen severity of the disease course or lessen adverse effects of infection. Alternatively, adding antimicrobial polyphenols such as those found in curcumin, cinnamon or green tea may also affect the course of *c. difficile* infection and outcome, as it is observed that in Indian populations that *c. difficile* infection is less common than in western countries (anecdotal observations); moreover, we do not assume that any differential antibiotic stewardship contributes to these observations [10].

Previously, it was demonstrated that treatment with Clindamycin or Piperacillin/Tazobactam weakened what we call colonization resistance that the protective enteric bacteria that inhibit overgrowth by *Clostridioides difficile* [10,19]. Moreover, changes in MG can be reflected in the urinary metabolome. Clearly, many MG-generated chemicals are the product of plant dietary polyphenols. In regard to teas an oral administration challenge should amplify MG-dependent changes in urinary small phenolic acid and other metabolites. Indeed, differential changes in murine urinary downstream polyphenol metabolites of green tea, namely, pyrogallol levels identified antibiotic treatment (clindamycin, piperacillin/tazobactam) previously associated with a weakening of colonization resistance to *Clostridioides difficile* and were affected by these antibiotic treatments. Furthermore, there are specific micro-organisms that are involved in EGCG biotransformation and metabolism, namely, *Clostridium orbiscindens*, *Enterobacter aerogenes*, *Raoultella planticola*, *Bifidobacterium longum* and *Eubacterium ramulus* in rodent studies [20] and in human subjects as well [21], suppressed levels were found of at least three small phenolic acids, namely, 3,2-hydroxyphenylpropionic acid (3,2-HPPA), 3,4-HPPA, 3,3-hydroxyphenylpropionic acid (3,3-HPPA and 3,4-hydroxyphenylpropionic acid [16,20]. Furthermore, 3-hydroxy-3-(3-hydroxyphenyl) propanoic acid (3,3-HPHPA). These small molecules are known to be at least partially derived by the gut microbiota through dietary flavonoids C-ring cleavage on the molecules [22,23].

3-hydroxy-3-(3-hydroxyphenyl) propanoic acid (3,3-HPHPA) was previously characterized and quantified in human urine, serum and cerebrospinal fluid samples [20] and was evaluated in rodent tissue samples [10]. This small phenolic acid is implicated in autism and other diseases and suggested to be caused by bacterial metabolism of precursor compounds [24].

In a qualitative survey of polyphenol cleavage generation from green tea in mice, and to a lesser extent in human urine (unpublished observations), was explored using a modification to a published mass spectrometry method [25]. Applying a targeted metabolomics method, 9 isomers, commercially available, were separated from 12 known isomers of HPHPA (See Figure 2). As a screening tool, the unequivocal separation of these metabolites was accomplished separated 9 of the available isomers from a mixture of standards using a Phenomenex Kinetex® F5 core shell technology column. Another isomer from a previous study was identified for the first time in mouse brains [25]. Previously, we studied how the addition by oral administration of polyphenol-rich green tea in mice affected the profiles of 3,3-HPHPA and 3,4 HPHPA Unpublished Obsersavions)and its other isomers in the mouse, which suggests the strong potential use of green tea as a metabolic tracer for many of these isomers. Furthermore, these cleavage products of polyphenols

demonstrate that bacterial co-metabolism is a key player in the production of bioactive molecules in mammals. This will extend microbiomic and metabolomic studies, particularly as applied to infectious disease and to the microbiota–gut–brain axis. Our novel approach supports identification, separation and quantification of currently available 3,3-HPHPA isomers and should be applicable to the remaining three commercially unavailable isomers, should they become available in the future [12].

**Figure 2.** Isomers (C9 H10 O4) with molecular weight 182.17 Da. (3-hydroxy-3-(3-hydroxyphenyl) propanoic acid (3,3-HPHPA), 3-hydroxy-2-(3-hydroxyphenyl)propionic acid (3OH-2,3-HPPA), dihydroxyhydrocinnamic acid (DHHCA), hydroxyphenyllactic acid (HPLA). * Currently available only by custom synthesis.

The intestinal microbiota, which consists of a diverse number of bacteria, archaea, viruses and even fungi [26], inhabits the gastrointestinal tracts of animals, including humans [10,27]. These microbial communities contribute to food and nutrient digestion and absorption and help in the development of the host immune system [26]. Recent rapid advances in DNA sequencing technology have allowed researchers to conduct in-depth studies of the structures of intestinal microbiota and their effects on host physiology. Previous studies have reported that intestinal microbial dysbiosis could be a risk factor for several diseases, including obesity, diabetes, atherosclerosis, neurodegenerative and mood disorders, immune system disorders, colon and hepatic cancers [28].

## 4. Oxidative Stress, Immunomodulatory and Anti-Inflammatory Properties

Polyphenolic phytochemicals are dietary antioxidants that have the ability to scavenge free radicals by donating hydrogen atom or electrons or by chelating metal cations [29,30]. ROS or RNS are free radicals and are involved in the oxidative damage of proteins, nucleic acids and lipids [31]. One could say that this is a paradoxical part of aerobic metabolism; however, oxidative stress appears to play a role in the pathobiogenesis of nearly every disease process [32]. However, the human body has a complex endogenous antioxidant defense system, such as the glutathione systems, glutathione peroxidase, and superoxide dismutase and catalase, which scavenge many free radicals. Interrupting these defenses can contribute to irreparable DNA damage, cellular membrane alterations, damage to cellular components and eventually leads to cellular death or apoptosis.

Polyphenols and their small phenolic compounds are secondary metabolites found most abundantly in plants, which also occur as bacterial transformations but not in animals, and thus explain why the aromatic amino acids are essential amino acids. However, since tyrosine can be converted from phenylalanine, tyrosine is not an essential amino acid in humans. However, without a metabolic tracer, it is difficult to determine what amount of tyrosine is bacteria-derived versus host converted. The intermediates are most often derived from the shikimate pathway (See Figure 3) These aromatic molecules have important roles, such as antioxidants, pigments, structural elements signaling factors and as defensins against pathogens, parasites, and infectious agents or from biotic and abiotic stress such as radiation. The phenolic acids and polyphenols are formed in higher plants and microorganisms from shikimic acid, which is a central metabolite, through an aldol-type condensation of phosphoenol-pyruvic acid and the glycolytic pathway via sugars and the pentose phosphate cycle, to produce 3-deoxy-D-arabino-heptulosonic acid 7-phosphate. The key branch-point on the production of phenolic compounds in plants is chorismic acid [33].

**Figure 3.** A variety of polyphenolic compounds, including caffeic acid, ferulic acid, vanillin, vanillic acid, and resveratrol are biosynthesized in plants through shikimate pathway and subsequent secondary metabolic pathways.

In microorganisms, these secondary metabolites are formed through the shikimate pathway (See Figure 3) to produce several aromatic amino acids, namely, L-tyrosine, L-phenylalanine and L-tryptophan [34], which are essential molecular building blocks for protein biosynthesis. In plants, these amino acids are crucial components for the synthesis of protein they also serve as precursors that are important for these diverse secondary metabolites and (SPMs) for the molecules we elucidate. Most notable are the aromatic phenolic compounds synthesized from L-Tyr, L-Phe such as HPHPA, cinnamoyl glycine, cinnamic acids, cinnamic esters, coumarins, phenylpropenes, flavonoids, isoflavonoids, neoflavonoids, stilbenes such as resveratrols, anthraquinones, chalcones and lignans [35]. The shikimic acid pathway involves chorismite enzyme catalyzed conversion of shikimic acid to chorismic acid, the enzyme catalyzed Claisen rearrangement of which gives pre-phenic acid. Prephenic acid, in a sequence of reactions, involving oxidative decarboxylation and transamination gives L-tyrosine, which is a precursor of various polyphenolic compounds that are formed as secondary metabolites in the gut microbiota [34,36–38].

Polyphenols, alone or as contained in whole foods and beverages, have been shown to prevent and alleviate oxidative stress-related metabolic disorders and damage [39] and some enzymatic mechanisms underlying these effects. The Shikimic acid pathway leading to the formation of L-phenylalanine and L-tryptophan in plants also undergo gut microbial metabolism to form various polyphenolic secondary metabolites. The gut microbiota can transform the ingested polyphenolic compounds, such as flavonoids (including anthocyanins, flavanones, and flavanols), to a plethora of polyphenol-based secondary metabolites. Although polyphenolic compounds exert positive outcome in health, some of the polyphenolic compounds may be involved in the expression of enzymes that mediate the pathways responsible for the onset of gut disorders, including colon cancer [40]. It has been well documented that alterations in inflammatory pathways are involved in development of gut disorders including colon cancers. When cytokine-stimulated human colonic fibroblasts were used to assess anti-inflammatory activity of dietary phenolic acids, some phenolic acids, including Gallic acid and Gentisic acid, demonstrated potential pro-inflammatory activity, whereas most others including Cinnamic acid, Vanillic acid, and Caffeic acid demonstrated anti-inflammatory activity [41].

Oxidative stress and inflammation largely drive proliferation, carcinogenesis and apoptotic processes. The apoptotic, cellular anti-proliferative, anti-inflammatory and anti-carcinogenic effects of polyphenols are best illustrated with the cinnamates, Cummins (free-radical sequestrating compounds) and has multiple mechanisms of action along with eugenol or 4-allyl-2-methoxyphenol all have been studied for ability to modulate cancer, diabetes, neurodegeneration and inflammation. Curcumin and resveratrol (3,5,4′-trihydroxy-trans-stilbene) both exert their effects, in part, by inhibiting NF-kB, TNF-a, IL-1b, IL-6, and COX-2 gene expression and downstream signaling [42]. Of these, eugenol is the quintessential compound in the group with many diverse uses to date. Eugenol and the cinnamates are present in widely diverse plant families including cloves (*Eugenia caryophyllata*), turmeric (*Curcuma longa*) [43], ginger (*Zingiber officinale*) and the bark leaves of cinnamon (*Cinnamomum verum*), among others [44,45]. Polyphenols such as curcumin have metal chelating properties, particularly for iron ($Fe^{3+}$ and $Fe^{2+}$ ions) and superoxide trapping activity [1].

Inflammation is part of the adaptive immune response, which can be triggered by deleterious stimuli, infection or injury, whether chronic or acute. Furthermore, oxidative stress and inflammation are mechanistically linked [1]. It is important to understand that free radical damage is inevitable, but there are many endogenous and protective antioxidant enzyme systems in healthy, euglycemic and younger individuals and damage accumulates with advancing age and under diabetic conditions [46]. In response to the inflammatory agents, the body induces many inflammatory molecules, such as cytokines and cytochrome oxygenase-2 (COX-2) enzyme. Several small polyphenolic acids, including sinapic acid, *p*-coumaric acid, cinnamic acid, vanillic acid, caffeic acid and ferulic acid, bind to the active site of COX-2 enzyme and thereby decrease the potentially neoplastic

prostanoids and prevent the onset of some cancers and gastro-intestinal disorders, such as colon cancer. These small molecules work to suppress inflammation and cytokine stress. On the other hand, some polyphenolic acids, such as gentisic acid or gallic acid, may be pro-inflammatory or even upregulate cytokine stress and signal transduction pathways that could lead to the upregulation of COX-2 [40].

Nevertheless, eugenol, curcumin and cinnamon have few side effects and tremendous anti-inflammatory and antioxidant effects. Curcumin has the same potential to be of medicinal value [47]. When eugenol's anti-inflammatory activity was explored in lipopolysaccharide-induced lung injury, decreased proinflammatory cytokines, such as TNF-$\alpha$ were noted [48], as was NF-κB-mediated pathway activation [42] and eugenol inhibits TNF-$\alpha$ and cyclooxygenase-2 (COX-2) expression [46] and also suppresses NF-κB-stimulating activation of macrophages due largely to TNF-$\alpha$ [48]. Moreover, ROS and RNS-mediated oxidative stress results in cellular, membrane and lipid peroxidation damage and increased COX-2, iNOS, and cytokine tumor necrosis factor $\alpha$ (TNF-$\alpha$) expression [48–50]. Eugenol suppresses proliferation of MCF-7 cells in a time and dose dependent manner [51,52] and has demonstrated antiproliferative action with its biphenyl (S)-6,60-dibromo-dehydrodieugenol derivative by initiating apoptosis. Thus, eugenol is a very interesting SPM, with diverse biological activities and very effective natural compound. Moreover, eugenol can biotransformed to many compounds such as vanillin through coniferyl alcohol and ferulic acid [53]. Furthermore, cinnamon promotes several bacterial species including *kkermansia*, *Bacteroides*, *Clostridium III*, *Psychrobacter* [54].

## 5. Antipyretic, Antiviral, Antifungal and Analgesic Properties

The antibacterial and antiviral pharmacology of eugenol have been known for over a century. Many who have had oral surgery may be familiar with the analgesic and antimicrobial bactericidal effects of eugenol, which has activity against a variety of strains of Gram-negative (*Pseudomonas aeruginosa*, *E. coli*, *Yersinia enterocolitica*, *Salmonella choleraesuis,* and even *Helicobacter pylori*) and Gram-positive (*Stapholococcus aureus*, *Streptococcus pneumonia*, *Enterococcus faecalis*, and *Streptococcus pyogenes*) bacterial strains [55]. It is suggested that the free hydroxyl group in the molecule imparts antimicrobial activity, and combinatorial approaches with eugenol and other antibiotics are more effective. However, it is not only bactericidal but virucidal activity that we observe as well. In that regard, the replication of the Herpes Simplex Virus (HSV) is also neutralized by neutralizing and inactivating this and other viral infections [56] through glycoprotein B that blocks HSV-1 and HSV-2 replication [57]; some studies suggest that eugenol inhibits viral DNA replication and can damage the outer envelope of newly synthesized virions [57].

Perhaps the most striking regulatory effect of polyphenols and phenolic acids is eugenol use in dentistry, where it is complexed with tooth fillers, used as an antiseptic/disinfectant and for its powerful analgesic properties [58,59]. Anyone who has used eugenol when suffering dry sockets, tooth pain or infection can attest to the striking relief and analgesic effect. This mechanism is linked to suppression of $Na^+$, $K^+$, and $Ca^{2+}$ voltage-dependent channels [60,61], particularly through high-voltage-activated $Ca^{2+}$ channel inhibition. These currents are active in both capsaicin-insensitive and capsaicin-sensitive dental primary afferent neurons, explaining why there is pain relief with this compound [61] (See Table 1).

**Table 1.** Polyphenol compounds and their physiologic role involving select colonizing bacteria species.

| Polyphenols | Sources | Chemical or Physiological Functions | Colonizing Bacteria |
|---|---|---|---|
| Curcuminoids | Dietary, Plants, such as: *Curcuma longa* | Shikimate pathway intermediate inhibits NF-kB, TNF-a, IL-1b, IL-6, and COX-2 gene expression, anti-inflammatory activity | *Lactobacilli* sp. *Prevotella* sp., Many uncharacterized species |
| (-)-epigallocatechin-3-*O*-gallate | Dietary, Plants, such as: Green tea (*Camellia sinensis)* | Shikimate pathway intermediate, chelating properties, anti-inflammatory and oxidative stress activity | *Clostridium orbiscindens, Enterobacter aerogenes, Raoultella planticola, Bifidobacterium longum, Eubacterium ramulus* in |
| (-)-epicatechin-3-*O*-gallate | Dietary, Plants, such as: green tea (*Camellia sinensis)* | Shikimate pathway intermediate, chelating properties, anti-inflammatory and oxidative stress activity | *Clostridium orbiscindens, Enterobacter aerogenes, Raoultella planticola, Bifidobacterium longum, Eubacterium ramulus* in |
| (-)-epigallocatechin | Dietary, Plants, such as: green tea (*Camellia sinensis)* | Shikimate pathway intermediate, chelating properties, anti-inflammatory and anti-oxidative stress activity | *Clostridium orbiscindens, Enterobacter aerogenes, Raoultella planticola, Bifidobacterium longum, Eubacterium ramulus* in |
| (-)-epicatechin | Dietary, Plants, such as: green tea (*Camellia sinensis)* | Shikimate pathway intermediate, chelating properties, anti-inflammatory and oanti-xidative stress activity | *Clostridium orbiscindens, Enterobacter aerogenes, Raoultella planticola, Bifidobacterium longum, Eubacterium ramulus* in |
| Cinnamon | Dietary, plants, such as: bark of *Cinnamomum versum* | Shikimate pathway intermediate, chelating properties, anti-inflammatory and anti-oxidative stress activity | *Enterococcus* spp. and *Lactobacillus* spp., *Campylobacter* spp. and *Enterococcus* spp. *kkermansia, Bacteroides, Clostridium III, Psychrobacter* |
| Eugenol | Dietary, Plants, such as: *Eugenia caryophyllata* | anti-inflammatory activity, decreased proinflammatory cytokines, such as TNF-α, NF-κB, synthesis of ferulic acid and other aromatic compounds via shikimate pathway through enzymes phenylalanine ammonia lyase; tyrosine ammonia lyase; S-adenosyl methionine (methyl donor). | Many uncharacterized species, *Staphylococcus aureus, Pseudomonas aeruginosa* |

## 6. Polyphenols in Obesity

Obesity is not only a chronic disease itself but is also a risk factor for various other chronic diseases. Obesity contributes to inflammation, oxidative stress, type 2 diabetes, cardiovascular disease, hypertension, and some types of cancer. It has been demonstrated that some food ingredients including polyphenols have anti-obesity effects. Powder from carob pods from an evergreen tree carob cultivated mainly in the Mediterranean region is used as a substitute for cocoa powder because of its color and flavor [62]. Carob pod powder (CPP) contains polyphenols and minerals such as calcium, phosphorus and potassium. The antioxidant activity is correlated with the polyphenol concentration in carob pod extracts [63]. It has been indicated by an in vivo study that carob pod polyphenols suppress the increases in adipose tissue weight and adipocyte hypertrophy in high fat diet-induced obesity model mice [64]. Feeding obese mice induced by a high-fat diet with CPP reduces the serum total cholesterol level and suppress lipid accumulation in hepatocytes caused by a high-fat diet. The in vitro mechanism studies using 3T3-L1 preadipocytes reveal that carob powder polyphenols suppress triglyceride accumulation in differentiated 3T3-L1 cells, which is corelated with deceased CEBPβ protein levels and PPARγ mRNA levels [64].

In the human body, there are two types of adipose tissues, including white adipose tissue (WAT) and brown adipose tissue (BAT). There is also visceral and adipose fat, with visceral being the most metabolically active. WAT stores extra energy as triacylglycerol, whereas BAT dissipates energy and releases it as heat [65]. Therefore, research in WAT browning, facilitating BAT activity and thermogenic functions of dietary polyphenols is a

great interest in treating obesity [66]. Various dietary factors including polyphenols fostering browning of WAT have been revealed to promote thermogenesis [67]. It has been demonstrated that apple polyphenols decrease adipose tissue mass [68]. When mice were used to study the underlying mechanisms, daily apple polyphenols consumption induced increased expression of brown/beige adipocyte selective genes UCP1, CIDEA, TBX1, CD137 [69].

## 7. Polyphenols in Heart Disease, Cancer, and Type 2 Diabetes

T2D is characterized by hyperglycemia, which can result from or contribute to defective insulin secretion, glucose intolerance and insulin resistance. Epidemiological and clinical evidence supports the idea that regular, moderate wine consumption (one glass or two per day) is associated with the decreased incidence of hypertension, T2D and cardiovascular disease [70]. Wine polyphenol constituents [71], such as those found in aged red wine and beer, particularly include resveratrol and trans-resveratrol. These compounds have some importance for aging, neurodegenerative diseases and cancer, including colon, basal cell, ovarian, and prostate carcinoma [70], in part because of their effect on certain enzymes that post-translationally modify patterns of histone protein acetylation [65,72]. Resveratrol and trans-resveratrol are stilbene derivatives also found in red wine are both of interest for the cardiovascular system and the brain, but holds promise for aging and may have properties other than those of antioxidants. Resveratrol in red wine is included among the two main types of polyphenol constituents, namely, flavanols and anthocyanidins (See Figure 1) [73]. These oxidized forms of flavanols are good metal chelators due to their vicinally located hydroxyl groups [1].

One quintessential example is resveratrol is a small molecule and natural polyphenol from various plants including grapes, cocoa, cranberries, strawberries, tomatoes, peanuts among others [74,75] and its isomers are believed to be the chief anti-aging compounds in red wine. Animal studies using rodent models reveal that resveratrol is effective in the treatment of obesity and T2D [76]. Polyphenols, such as resveratrol, tea catechins and curcumin, are increasingly recognized for their role in the regulation of lipid metabolism in the host, perhaps through the microbiota [77]. These improve glucose hemostasis, lipid profiles, metabolic efficiency and body weight loss [76]. Resveratrol causes phosphorylation of AMPK, and thus improves its activity while decreasing the expression of the enzymes involved in lipogenesis and improving insulin-mediated glucose uptake [78].

Curcumin and resveratrol both may improve insulin sensitivity cardiovascular function and are suspected to play a role in the so-called French paradox [79,80]. Aging is associated with arterial stiffening, systolic blood pressure and heart disease. Interestingly, Trimethylamine N-Oxide (TMANO), the gut microbiome-derived metabolite, was found to induces aortic stiffening, increase systolic blood pressure in rodents and humans with advancing age [81]. The role of the microbiota is now part of a heart shunt in the MGB axis, as it relates to a vegetarian and polyphenol-rich diet, which provides health-promoting phytochemicals and phytonutrients that are beneficial for both the heart and the brain through the gut microbiota [80].

Polyphenols are not the only protective compounds in a vegetable-rich diet or in wine because other putative cardioprotective factors may involve folate as a contributor to the effect of wine. The fundamental mechanisms underlying the beneficial effects of polyphenols in relation to metabolic disorders and the gut microbiota in murine models show polyphenols ameliorate effects of metabolic disorders by locally alleviating intestinal oxidative stress, inflammation and improve intestinal barrier function by modulating microbial colonization with short-chain fatty acids producing bacteria [82,83]. Furthermore, it was found that the presence of tight junctions in the digestive tract of rodents can be improved, thereby implicating its role in strengthening the gastrointestinal barrier [84]. When the anti-obesity effects of cinnamon (*Cinnamomum Zeylanicum*) are studied using diet-induced obese male adult Zebrafishes, feeding them a cinnamon-treated diet reduces BMI, blood glucose levels and lipid levels in the liver, which are associated with decreased

expression of genes involved in adipogenesis such as PPAR family genes that are induced by a high-fat diet [85].

Curcumin is widely used in Asia as a culinary ingredient in food recipes. Curcumin, [1,7-bis(4-hydroxy-3-methoxyphenyl)-1,6-heptadiene-3,5-dione], is the lipophilic polyphenol component extracted from rhizome of turmeric (*Curcuma longa*). It is donor of electrons to reduce reactive oxygen species (ROS), therefore functions as an antioxidant [86]. Since it is almost insoluble in water, its anti-inflammatory, antitumor, antimicrobial, and antiviral functions may be carried out by its metabolites absorbed into the circulation. It may also function through regulation of the species of microbes in the gastrointestinal tract. When mice with colon cancer are fed a diet containing curcumin, curcumin administration eliminates or reduces the colon tumor burden, which is associated with increasing *Lactobacilli* and reducing *Coriobacterales* [87,88] In addition, curcumin treatment decreases microbial abundance of cancer-related species, such as *Prevotella*, that were found to be greater in the stool of colorectal cancer patients [89]. It is well known that increased consumption of plant-derived foods is inversely correlated with T2D. A recent finding suggests that in T2D, polyphenols from fruits, vegetables and plants, which is related to dietary polyphenol intake can significantly help in the maintenance of glycaemic control and diabetes prevention, as well as the aggregation of amyloid fibrils in confirmational diseases and downstream toxicity [90]. Moreover, vitamins, such as thimine (vitamin B1), which are also derived from the same sources, could work in concert with polyphenolic compounds to prevent or alleviate T2D in part [91].

## 8. Epigenetic Modifications Targeted by Polyphenols

Epigenetic modifications are the modifications to the genome, rather than changes in the DNA sequences, that can cause alterations of gene expression. These modifications include DNA methylation, histone protein modifications (acetylation, phosphorylation, and methylation), and mircoRNA (miRNAs) [92]. The changes in these epigenetic modifications can be induced by environmental factors including dietary factors and lifestyle factors, and are subsequently involved in the development of chronic diseases such as obesity, T2D, cardiovascular diseases, and various cancers. The mechanisms of some of the chemical or physiological functions such as anti-inflammatory, anti-oxidant and anti-cancer properties of polyphenols can include epigenetic modifications [93]. Polyphenols can reserve adverse epigenetic modifications by changing DNA methylation, histone protein modifications and miRNA levels [93]. DNA methylation is catalyzed by DNA methyltransferases (DNMTs) by transferring a methyl group from S-adenosyl-methionine (SAM) to a cytosine of CpG dinucleotides [92]. The acetylation levels of histone proteins are controlled by histone acetyltransferases (HATs) and histone deacetylases (HDACs). HATs promote histone acetylation and result in a more relaxed chromatin structure that mostly favors transcriptional activation, whereas HDACs remove acetyl groups and consequently suppress gene expression [94]. It has been indicated that curcumin acts as an epigenetic modulator to inhibit DNMTs, regulate HATs and HDACs, and regulate miRNAs in addition to binding to DNA and interacting with transcription factors [93]. We proposed early on that histone modification such as phosphorylated histone H3 was important in early Alzheimer disease pathogenesis [95].

## 9. Conclusions

Our conclusions and future direction point to a better understanding of the second genome within us and co-metabolism within all animals involving the gut microbiota. An old adage says, "an apple a day keeps the doctor away", and perhaps this is due to the diverse polyphenols within apples that explain this anecdote. In that regard, we call for a new physiology, which involves co-metabolism within the GM and exploring this metabolic aspect. This is now entirely possible with the advancements in deep and whole ge-



nome sequencing [72,96]. Moreover, the mutually beneficial metabolic relationship between the host and its resident gut microbiota has been widely described [10,12,84,97]. The bacterial-derived dietary products and metabolites from gut commensal micro-organisms are largely useful for the host and our overall health and it is arguable the polyphenols and phenolic acids largely derived from the Gram-negative enteric species that are most important of these. Co-metabolism, which occurs between the microbiota and host systems and some of these same microbes, can control integral segments of our overall health, neurobiology, mood, biochemistry and may even be integral in the future control of disease.

**Author Contributions:** All authors contributed to this work, the main writing was done by the first 2 authors. All authors have read and agreed to the published version of the manuscript.

**Funding:** This research received no external funding.

**Institutional Review Board Statement:** Not applicable.

**Informed Consent Statement:** Not applicable.

**Data Availability Statement:** Not applicable.

**Acknowledgments:** The author wishes to thank William Francis of Westlake, Ohio, for introducing the amazing metabolic and analgesic properties of the phenolic acid eugenol as a dental treatment.

**Conflicts of Interest:** The authors declare no conflict of interest.

**Abbreviations**

(MG) Microbiota–Gut, (MGB) Microbiota–Gut–Brain, (EGCG) (-)-epigallocatechin-3-*O*-gallate, (ECG) (-)-epicatechin-3-*O*-gallate, (EGC) (-)-epigallocatechin, (EC) (-)-epicatechin, (SPM) small phenolic acids and molecules, (AD) Alzheimer's disease, (PD) Parkinson's disease, (VD) vegan diet, (MD) meat-rich diet, (ROS) Reactive Oxygen Species, (RNS) Reactive Nitrogen Species, acid, (HSV) Herpes Simplex Virus, (HPPA) Hydroxyphenylpropionic acid, (HPHPA) 3-hydroxy-3-(3-hydroxyphenyl)propanoic acid, (miRNAs) mircoRNAs, (C. Difficile) *Clostridioides difficile*, type 2 diabetes (T2D).

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
