# Peer review of "Polyphenols and Small Phenolic Acids as Cellular Metabolic Regulators"

_cimb, doi:10.3390/cimb44090285_

Round 1
Reviewer 1 Report
The review entitled “Polyphenols and Small Phenolic Acids as Cellular Metabolic Regulators” aims to “….explore the role of polyphenols and phenolic compounds as metabolic or intrinsic biochemistry regulators in the context of the microbiota-gut-target organ axis…..”
While the topic is surely really interesting, the manuscript has some important issues to be taken into account before publication.
In general all the paper has to be formally revised, too many paragraphs are not clear, too vague, or repetitive. Just as example, but far to be exhaustive:
- - In introduction, paragraph 58-60 is exactly repeated as paragraph 60-62, and after again in 80-82
- - Sections 4 and 5, appear as a description of some poliphenols (mainly curcumin and eugenol) without almost any discussions concerning the interactions with microbiota (declared aim of the work)
- - Too often this work appears not as a review, but as a research paper of the author reporting their thoughts, considerations, and with references to their works sometimes declared as “unpublished observations” (a review is supposed to report the state of the art, and not references impossible to check).
- - Section 6 is a scholastic introduction to a topic without any real data
- - Bibliography is not updated at all, it needs to be completely revised. Just in 2021 and so far in 2022 exist more than 1000 published papers for year, with polyphenols and microbiota as keywords, and practically none of these is here reported.
- - References sections has to be totally formatted, and numbers occuring in the text should be in the correct order (just in first page there is a jump from 4 to 87, and then 26…). Furthermore some references are duplicated such as 23 and 86, or 68,69 and 71.
Author Response
We had added all requested changes attached

Reviewer 2 Report
The manuscript entitled "Polyphenols and Small Phenolic Acids as Cellular Metabolic Regulators" summarizes the most important research on the role of polyphenols as regulators in the context of the microbiota-gut-organ axis, being very interesting. I recommend its publication with a few observations:
- Medicinal plants named in Latin (Curcuma longa, Cinnamomum verum) are written in italics.
- Creating a table with the most important polyphenols (their source, mechanism, colonizing bacteria, actions, etc.) discussed in the paper to help systematize the material.
- To be corrected: anti-inflammatory or paragraphs written not in accordance with the requirements
- Please give an example with polyphenolic compounds which may be involved in the expression of enzymes that mediate the pathways responsible for the occurrence of intestinal disorders, including colon cancer
Author Response
We have added and responded to all requested critiques

Round 2
Reviewer 1 Report
i have reviewed again the manuscript, and the style has been indeed improved. It's still necessary a better formatting of Table 1 (too many styles, and some mistake in use/not use of italic), after which it can be considered for publication.